# Open design of a reproducible videogame controller for MRI and MEG

**Yann Harel**[1,2☯]*, **André Cyr**[1,3☯], **Julie Boyle**[1,2,3], **Basile Pinsard**[1,3], **Jeremy Bernard**[4], **Marie-France Fourcade**[4], **Himanshu Aggarwal**[5], **Ana Fernanda Ponce**[5], **Bertrand Thirion**[5], **Karim Jerbi**[2,3,6,7], **Pierre Bellec**[1,2,3]

**1** Centre de Recherche de l'Institut Universitaire de Gériatrie de Montréal, Montréal, Canada, **2** Psychology department, University of Montréal, Montréal, Canada, **3** Unité de Neuroimagerie Fonctionnelle, Centre de Recherche de l'Institut Universitaire de Gériatrie de Montréal, Montréal, Canada, **4** Neurospin, CEA, Gif-sur-Yvette, France, **5** Inria, CEA, Université Paris-Saclay, Palaiseau, France, **6** MILA, Montréal, Canada, **7** MEG Imaging Center, University of Montréal, Montréal, Canada

☯ These authors contributed equally to this work.
* yann.harel@umontreal.ca

**Data Availability Statement:** All the data used in this study are made available as part of the Courtois-Neuromod project. Access can be obtained by filling a short form at https://www.cneuromod.ca/access/access/. The preprocessed

## Abstract

Videogames are emerging as a promising experimental paradigm in neuroimaging. Acquiring gameplay in a scanner remains challenging due to the lack of a scanner-compatible videogame controller that provides a similar experience to standard, commercial devices. In this paper, we introduce a videogame controller designed for use in the functional magnetic resonance imaging as well as magnetoencephalography. The controller is made exclusively of 3D-printed and commercially available parts. We evaluated the quality of our controller by comparing it to a non-MRI compatible controller that was kept outside the scanner. The comparison of response latencies showed reliable button press accuracies of adequate precision. Comparison of the subjects' motion during fMRI recordings of various tasks showed that the use of our controller did not increase the amount of motion produced compared to a regular MR compatible button press box. Motion levels during an ecological videogame task were of moderate amplitude. In addition, we found that the controller only had marginal effect on temporal SNR in fMRI, as well as on covariance between sensors in MEG, as expected due to the use of non-magnetic building materials. Finally, the reproducibility of the controller was demonstrated by having team members who were not involved in the design build a reproduction using only the documentation. This new videogame controller opens new avenues for ecological tasks in fMRI, including challenging videogames and more generally tasks with complex responses. The detailed controller documentation and build instructions are released under an Open Source Hardware license to increase accessibility, and reproducibility and enable the neuroimaging research community to improve or modify the controller for future experiments.

## Introduction

Videogames offer a wide range of rich and dynamic environments which open exciting avenues for research in cognitive neuroscience [1]. Major neuroimaging techniques such as

data used to generate the figure is available in the Supplementary Information.

**Funding:** This project was supported by the Courtois Fondation (https://www.charitydata.ca/charity/fondation-courtois/850271289RR0001/), grant awarded to PB for the Courtois NeuroMod Project 2018-2024 (https://www.cneuromod.ca/). PB is a senior fellow ("chercheur boursier senior") of the "Fonds de recherche du Québec - Santé" (https://frq.gouv.qc.ca/). KJ is supported by funding from the Canada Research Chairs (950-232368; https://www.chairs-chaires.gc.ca/home-accueil-eng.aspx) program and a Discovery Grant from the Natural Sciences and Engineering Research Council of Canada (2021-03426; https://www.nserc-crsng.gc.ca/index_eng.asp). The reproduction of the controller by the MIND team, Inria, France was supported by funds from the European Union's Horizon 2020 Framework Programme for Research and Innovation (https://research-and-innovation.ec.europa.eu/funding/funding-opportunities/funding-programmes-and-open-calls/horizon-2020_en) under the Specific Grant Agreement No. 945539 (Human Brain Project SGA3) awarded to BT and through the joint Inria "NeuroMind" team grant to PB, KJ, BT and Alexandre Gramfort. The funders had no role in study design, data collection and analysis, decision to publish, or preparation of the manuscript.

**Competing interests:** The authors have declared that no competing interests exist.

functional magnetic resonance imaging (fMRI) and magnetoencephalography (MEG) adversely interact with any electronic equipment, which precludes the use of traditional videogame controllers during neuroimaging sessions. This paper reports on a videogame controller compatible with fMRI and MEG systems, developed by the team of the Courtois project on neuronal modeling (CNeuroMod). We evaluated the performance of this new device by comparing it to a regular, commercially available, videogame controller. We also assessed possible interactions between the controller and the quality of fMRI and MEG data acquisitions. In the spirit of open science, we released publicly the plans and documentation for building this controller, and confirmed that an independent team was able to build a reproduction based on the plans and documentation.

Neuroimaging videogames studies have examined how brain networks interact to produce a variety of complex behaviors such as aggression [2–5], learning [6, 7], expertise [8, 9] and flow [10]. Videogames often recruit many complementary abilities at once, e.g. spatial navigation and decision making in first-person shooters, thus probing the integration of diverse cognitive domains within a single environment [11]. Brain activity during videogame play is actively being studied with EEG (e.g. [12]) and fNIRS (e.g. [13]), two neuroimaging techniques that can be used with a traditional videogame setup, but only a handful of studies have attempted to use fMRI or MEG (e.g. [14]) to acquire high-quality functional brain data. Experiments conducted in MRI and MEG scanners need compatible response devices that do not create major acquisition artifacts or safety concerns [15]. These requirements prevent the use of commercially available controllers in research conducted using fMRI and MEG. Although possible to shield electronics [16], such a procedure only partially alleviates concerns on signal quality and possible safety concerns in an MRI environment. Videogames collected during fMRI sessions have relied on non-ecological MRI-compatible response devices such as button pads [2, 4–6, 17, 18], joysticks [19, 20], trackwheel mouse [18], and data gloves [6, 8]. These devices do not necessarily match those normally used to play the corresponding videogames, for example a button pad instead of a gamepad [17], or a trackball [18] instead of a mouse. Atypical devices may negatively impact the experience and performance of players in a brain scanner, and severely limit the type of games which can effectively be played within a scanning environment.

A commercial, fully MRI-compatible handheld videogame controller was recently proposed by Current Designs Inc., Philadelphia, PA, USA, requiring the separate acquisition of an electronics interface and a fiber optics cord extension, resulting in significant costs. An alternative approach to commercial solutions for research has however emerged: Open Source Hardware (OSH), made possible by the increasing accessibility of 3D printing, engineering parts, and modular micro-electronics [21]. Inspired by the Free and Open Source Software philosophy, OSH refers to technological artifacts for which the design files, documentation, and required software are freely available under an open-source license specifically dedicated to sharing hardware products. This approach provides many benefits, in line with the principles of open science: promoting accessibility and reproducibility, significantly reducing construction costs, and allowing for iterative design and customization.

The aim of this study was to develop and validate a videogame controller that is both MRI- and MEG-compatible and can be reproduced by other neuroimaging research teams. This was achieved by making detailed documentation and design files accessible to allow for 3D printing and assembly. The controller was evaluated by comparing its responsiveness to a reference controller used in a mock scanner, in terms of button press and release latencies in a cued-response experiment. We also checked the impact of the CNeuroMod controller usage on subject motion, using frame displacement metrics derived from fMRI acquisitions. Finally, we acquired phantom MRI scans as well as MEG empty room data to determine if any change in

background noise could be attributed to the controller's presence in the scanning environment.

## Methods

### Controller

The documentation for building the controller is provided in the Appendix of this article and can also be found at https://controller-doc.readthedocs.io/en/latest/. It contains a Bill of Materials (BOM) of the commercial parts necessary to build the controller, detailed assembly instructions, source code for the microcontroller, and a download section for the 3D models and Gerber files. The 3D model files are available either as .STL files, which can be readily sent to a 3D printer, or as. STEP files for manipulation and modification using Computer-Aided Design (CAD) software. Gerber files describe the different layers of the Printed Circuit Board (PCB) that need to be sent to a manufacturer for production. The buttons layout of the controller is presented in Fig 1.

**Design choices.** The design of handheld videogame controllers has been heavily influenced by the early model of the Super Nintendo Entertainment System (SNES™, Nintendo). This controller features two sets of buttons: one for the left hand which consists of four arrow-like keys intended for spatial navigation, and another set of four buttons for the right hand offering various types of game-specific actions. The combination of these eight buttons is sufficient for players to explore rich action spaces and covers the basic requirements of a vast array of videogame titles, especially retro videogames (that is, games released on consoles of the generations preceding the PlayStation™ and XBox™ systems).

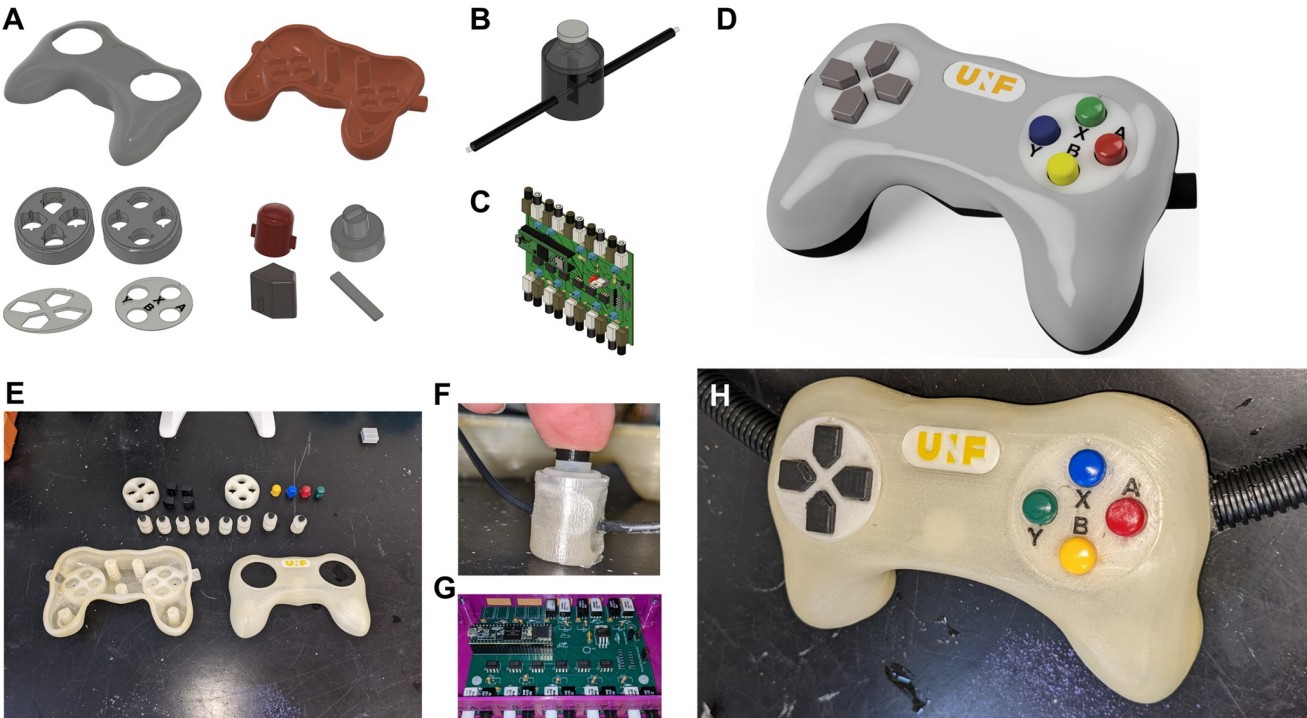

**Fig 1. Models and pictures of the different controller parts. A** Split view of the various 3D-printed parts **B** 3D models of the assembled switch part. **C** 3D model of the PCB. **D** 3D model of the assembled controller. **E-H** Real pictures of the corresponding elements as assembled at our facility.

The controller was designed with the six following specific criteria in mind. Accordingly, the handheld part of the controller must:

1. be built using a 3D printer and commercially available parts in order for it to be easily reproducible,

2. be entirely void of electronics and ferro-magnetic components to ensure MRI- and MEG-compatibility,

3. reliably register all button presses with a temporal precision comparable to that of the reference controller,

4. mimic the feeling and comfort of a reference commercial videogame controller, the Super NES controller, a device held with both hands each manipulating a set of 4 buttons using the thumbs,

5. allow users to comfortably play while lying down (position as imposed by MRI scanner and possible in some MEG scanners),

6. be comfortable to use for individuals with different hand sizes.

**Controller system.**   The device is made up of two parts linked by a bundle of fiber optics (Fig 2): the first is the actual handheld controller, and the second is a printed circuit board (PCB) that allows communication between the controller and a stimulation computer. The handheld controller and fiber optics bundle are MRI and MEG-compatible, and as such can be used in the MRI room or MEG chamber while the PCB contains unshielded electronics and must remain in the control room. Our pilot controller featured a 10m long fiber optics cable, which was sufficient to connect the controller in the MRI room with the PCB in the control room.

**Software.**   A Teensy 3.5 (Teensy®) microcontroller board was programmed to parse the optical signals and manage communication between the PCB and a stimulation computer. Programming was done in the Arduino Integrated Development Environment (IDE; Arduino) using the Teensyduino add-on (https://www.pjrc.com/teensy/teensyduino.html), which allows programming Teensy family boards using most of the Arduino libraries. This software must be uploaded to the Teensy board located atop the PCB.

## Task design

The datasets used in this study come from the Courtois NeuroMod databank (https://www.cneuromod.ca/gallery/datasets/). For the purpose of this project, we only utilized data from 4 subjects of 6 participants available in the databank, specifically the 4 subjects that performed the gamepad task. The code used for data analysis is available at https://doi.org/10.5281/zenodo.7847577 (or on GitHub https://github.com/courtois-neuromod/controller_validation).

**Subjects.**   Four participants (2 female) were scanned while they performed a variety of tasks in the MEG and MRI scanners. All four subjects were right-handed and had normal hearing for their age, ages ranged between 31 and 47 at first data set acquisitions in 2018 (i.e. *hcptrt* dataset), and the timespan between first and last dataset acquisition (i.e. *gamepad* dataset, see *Control datasets* section) was approximately 2 years and 7 months. Subjects were recruited as part of the Courtois NeuroMod project (CNeuroMod), a dense fMRI database that is being collected on 6 subjects using both naturalistic and controlled stimuli from a broad range of cognitive domains and the main motivator for building the controller. Although the

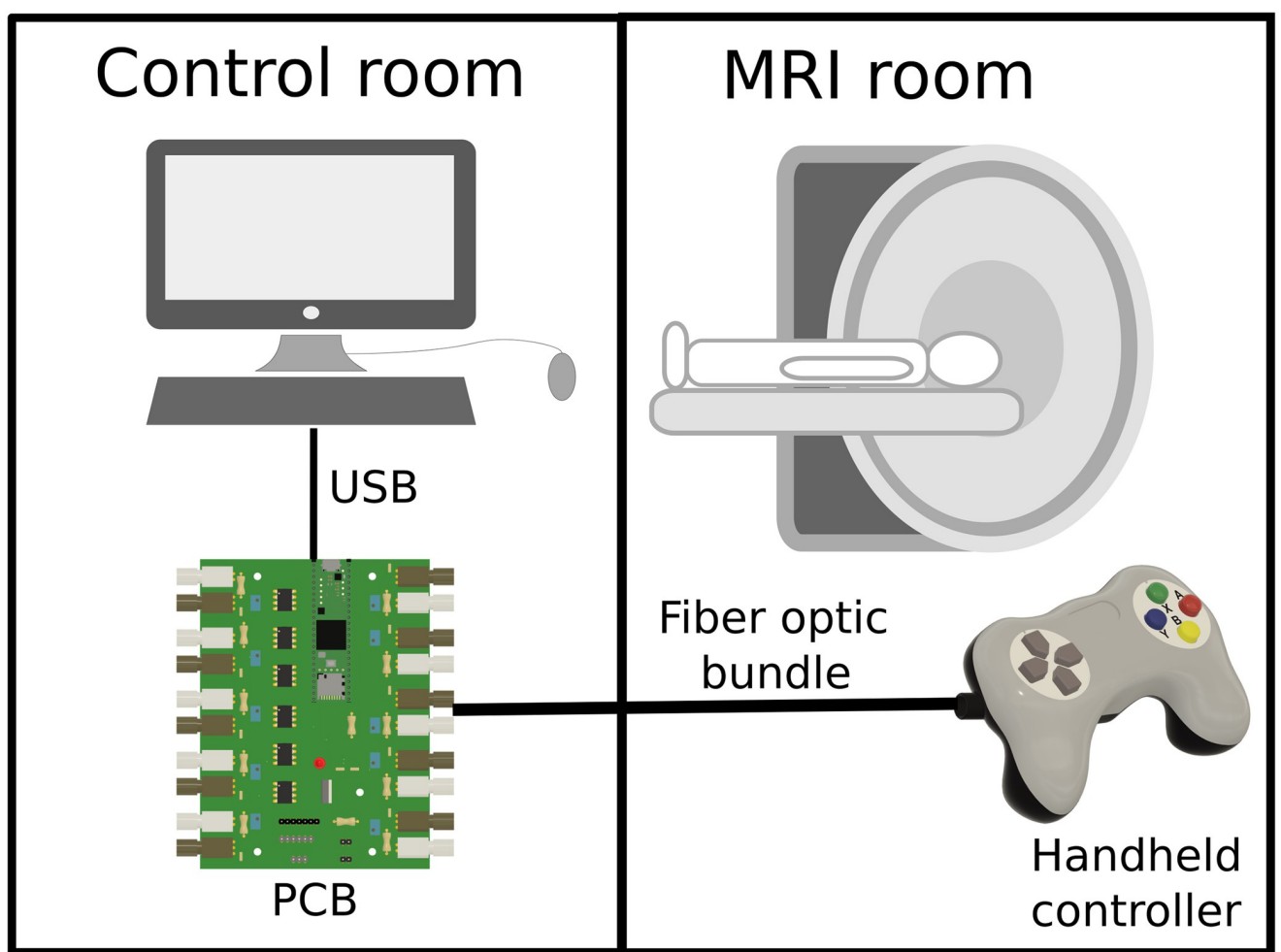

**Fig 2. Schematics of the two main parts of the controller.** The PCB is installed in the control room, while the handheld part is MRI-compatible and can be used in the MRI scanning room.

participants of the *gamepad* task do not properly reflect the population of videogame players, we expected their behavior to be illustrative of the controllers capability as the task didn't include videogame components per se.

**Ethics statement.** The study was approved by the "Comité d'éthique de la recherche vieillissement neuroimagerie" of the CIUSSS du Centre-Sud-de-l'île-de-Montréal (CIUSSS-CSMTL) under the number CER VN 18-19-22 and all participants provided written consent.

**Utility ratings.** A subjective evaluation of the controller in accordance with ISO-9241 on device comfort was assessed via a questionnaire of 8 items answered on a 5-points Likert scale, adapted from [22]. For the items smoothness during operation, general comfort and overall useability, higher scores indicate better fitness. For the items finger and wrist fatigue, lower scores indicate better fitness. For all other items, better fitness is indicated by scores around 3. All subjects of the CNeuroMod sample answered the questionnaire after having completed the totality of the validation task.

**MRI acquisition.** fMRI data used in this paper were collected as part of the CNeuroMod project using a 3T Siemens Prima Fit MR scanner, a 64-channel head coil, and an accelerated

simultaneous multi-slice imaging sequence [23]. The spatial resolution was 2mm isotropic, and the TR was 1.49s. All subjects wore custom CaseForge [24] head cases to minimize motion. Visual stimuli were projected onto a screen via a waveguide and presented to participants on a mirror mounted on the head coil, and sound was delivered using S15 Sensimetrics headphone inserts. Additionally an anatomical scan was collected using a T1-weighted MPRAGE 3D sagittal sequence, (TR = 2.4 s, TE = 2.2 ms, flip angle = 8 deg, voxel size = 0.8 mm). fMRI and anatomical data were preprocessed using the fMRIprep pipeline [25], using the "long term support" version 20.2.3. For a more detailed description of the MRI or fMRI sequences and set up visit the CNeuroMod'st documentation page: (www.docs.cneuromod.ca/en).

**Gamepad task.** We set out to evaluate the reliability of button responses of our custom controller by comparing it to that of a commercially available, non-MRI-compatible controller. Thus, this experiment consisted of a basic cued-response task implemented in PsychoPy [21] that was administered in two setups in a pseudo-randomized fashion: either in the MRI scanner using our homemade MRI-compatible controller, or in a mock scanner (i.e. a replica of a MRI scanner that does not contain a magnet) equipped with a generic commercial controller modeled after the classic SNES™ controller.

In the MRI setup our MRI-compatible controller was installed in the MRI scanner and connected through a waveguide to the controller interface, located in the control room. The controller interface was in turn connected to the stimulation computer via USB to register and log key presses during the task. The remaining set up was exactly as described above in the section on fMRI acquisition (Fig 3).

In the mock setup we used an MRI simulator made by Psychology Software Tools (https://pstnet.com/products/mri-simulator/), and used an in-house 3D printed replica of the Siemens 64 channel head coil used in the MRI. An in-house built mirror mount replicating the Siemen's mirror attached to the head coil was used to present images to the participants. Instead of a projection, visual stimuli were presented using a small monitor placed within the mock bore, closely replicating the manner in which visual stimuli are presented in the MR scanner. A replica of the SNES™ controller (iNNEXT Retro USB Super Classic Controller, iNNEXT) was installed inside the mock scanner, and directly connected to the stimulation computer via USB (Fig 3).

Participants were instructed they would have to perform a cued-response task, in which, using visual cues, they would have to press one of the 8 buttons present on either the MRI or SNES controllers, for a short or long period of time. Short presses were indicated by the

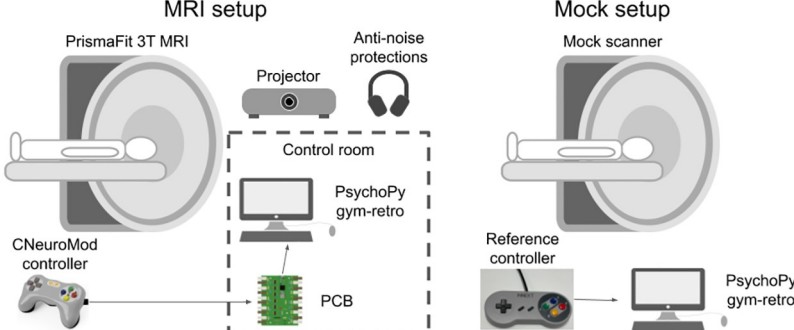

**Fig 3. Schematics of the MRI and Mock setups.** The mock setup closely replicated the MRI setup while allowing the use of a commercial controller instead of our MRI-compatible controller. Icons obtained under CC license at https://thenounproject.com/.

presence of a yellow dot prior during a trial and long presses were indicated by a green bar during a trial (Fig 4). In short trials, a yellow dot preceding the cue instructed the participants to press and release the cued button as fast as possible (cue displayed for 300ms), while in long trials participants were instructed to hold the cued button as long as the green bar was present on the screen (cue was displayed for a duration between 1 and 3 seconds with a gaussian jitter). Participants performed the tasks in 4 sessions, and during each session performed the task in both the MRI and mock setups (2 runs each). In a run, the subject performed 5 repetitions of each of the 8 buttons in the two conditions (short and long) for a total of 80 trials per run (5*8*2). Each participant performed 8 runs per setup (Mock and MRI) split in 4 sessions, for a total of 1 280 trials (80*8*2).

**Control datasets.** To illustrate the values of keypress durations that naturally occur when playing action platformer videogames, and use them as reference to interpret the results of our *gamepad* task, we computed the durations observed in the training phase of the *shinobi* dataset. The data for the *shinobi_training* dataset were obtained using the commercial controller (i.e non-MRI controller that was also part of our Mock scan condition of the *gamepad* task). This dataset comprised the same 4 subjects that performed the *gamepad* and *shinobi* tasks, and totals 1 165 trials across the 3 selected levels of Shinobi III: Return of the Ninja Master [26]. The game was played and recorded at 60 frames per second, allowing us to compute keypress durations with a precision of 16ms.

In order to estimate the amount of motion induced by our controller both in controlled and naturalistic conditions, we compared the motion (quantified by Framewise Displacement, FD) observed during the fMRI acquisitions of the *gamepad* and the *shinobi* datasets respectively with other datasets obtained on the same 4 subjects, chosen as a reference because they did not involve the controller. One exception is the shinobi dataset as sub-03 did not participate, and thus we only included data from 3 of the 4 subjects. The control datasets (*friends*, *hcptrt-gamble*, *hcptrt-motor* and *hcptrt-rest*) were selected to reflect the variety of motion amounts one can expect during conventional active and passive tasks. We used around 38h of fMRI data from the *friends* dataset during which the subjects were watching seasons 1 up to 4 of the Friends TV show [27]. Given the engaging nature of the stimuli and the passive nature of the task, we expected the associated FD values for this dataset to indicate relatively low motion [28].

We also used data from the *hcptrt* dataset, namely the resting state, gambling and motor tasks. All tasks in the *hcptrt* dataset were designed by the team at the Human Connectome

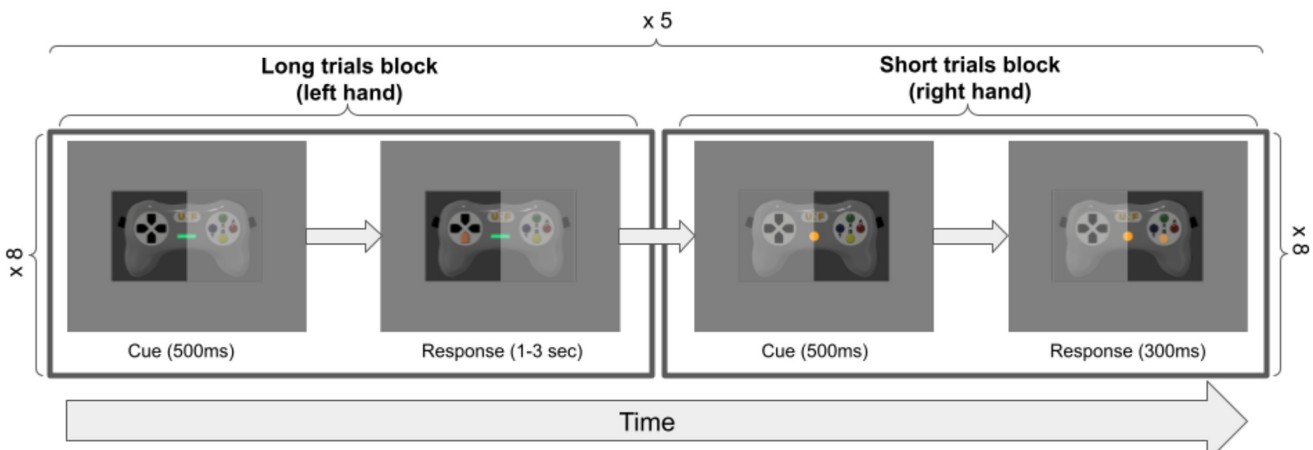

**Fig 4. Illustration of the unfolding of the simple cued-response task.**

Project [29] and each task was repeated up to 15 times. The resting state task was repeated 4 or 5 times per subject and consisted of staring at a cross (no response). The gambling task was repeated between 15 and 17 times per subject; using a standard MRI response box subjects were instructed to guess if they thought a hidden number (represented by a "?") was above or below 5.

The motor task was repeated between 15 and 18 times per subject. In this task, participants were presented a visual cue and were asked move a body part (ie. tongue, fingers, toes, hand, feet) in response to that cue. We expected these three datasets to reflect relatively low, moderate and high levels of motion for the resting, gambling and motor tasks respectively.

Lastly we used fMRI data from 3 of the participants in the *shinobi* dataset, which consists of about 10h (approximately 56 runs) of fMRI data per subject. During the shinobi task, subjects played three selected levels of Shinobi III: Return Of The Ninja Master [26], a retro videogame released on Sega Genesis™, using our custom videogame controller. We expected this last dataset to reflect the amount of motion typically observed during fMRI acquisitions involving videogame play with the CNeuroMod controller.

For more information on each of the control datasets, see the Courtois NeuroMod's Project documentation page: (https://docs.cneuromod.ca/en/latest/).

**fMRI phantom recordings.**   Thirty runs of 60 volumes were acquired using a fBIRN gel phantom with the custom controller in the scanner, and 30 more without the controller, all following the imaging sequence described in the *MRI acquisition* section. Each run lasted about 2 minutes, and a total of 60 runs were acquired on a single day.

**MEG empty room recordings.**   Empty room noise recordings were acquired as part of the routine procedure for MEG acquisitions. The data was collected on separate days with a MEG CTF-275 (with 270 functioning channels) while no participant was inside the MEG shielded room. In each recording, 35 seconds of signal were acquired at a sampling frequency of 1200Hz. No offset removal nor filters were used during these acquisitions. We used the empty room recordings acquired between 02/04/2019 and 09/06/2021 (N = 80), and split them into 4 subsets of 20 consecutive recordings. These subsets, named T1, T2, T3 and T4 include data acquired before the installation of our custom controller (T1, T2 and T3), and data acquired after the installation of our controller at the MEG facility (T4). Each of these subsets contained recordings spanning over between 1 and 3 and a half months.

## Analyses

**Response latencies (custom controller VS SNES controller).**   In order to verify that our controller had the same performance capacity as the commercial controller, we tested the hypothesis of a greater response latency in the MRI (using our controller) than in the mock scanner (using the commercial reference controller). We removed: (a) trials for which no button press of the instructed key was recorded; (b) trials with negative reaction times and durations, which correspond to trials that were anticipated by the subject due to the task's pseudo-random sequence; and (c) trials for which the press timing was recorded but no timing for the release was recorded due to an overlap with the subsequent trial. Additionally, we removed (d) trials from the "long-press" condition for which the press duration was below 1s, as these trials might have been mistakenly performed as "short-press" trials. In total, we removed 71 trials of the mock condition and 50 trials of the MRI condition (Table 1). Statistical analyses were conducted on pooled reaction times, and on press duration for long and short trials separately. One-tailed Kolmogorov-Smirnov tests for independent samples with FDR (Benjamini-Hochberg) correction for multiple comparisons were conducted between the two setups. Effects size were estimated using Cohen's D.

**Table 1. Number of removed trials at each step of the preprocessing pipeline.**

| | | A | B | X | Y | Up | Down | Left | Right | Total |
|---|---|---|---|---|---|---|---|---|---|---|
| Mistakes | Mock | 3 | 3 | 8 | 4 | 9 | 7 | 6 | 7 | 47 |
| | MRI | 6 | 3 | 0 | 3 | 6 | 3 | 3 | 2 | 26 |
| Negative RTs | Mock | 0 | 2 | 1 | 0 | 0 | 0 | 0 | 0 | 3 |
| | MRI | 0 | 1 | 0 | 0 | 0 | 1 | 0 | 0 | 2 |
| Negative durations | Mock | 0 | 0 | 0 | 0 | 0 | 0 | 0 | 0 | 0 |
| | MRI | 0 | 0 | 0 | 0 | 1 | 0 | 0 | 0 | 1 |
| Delayed trials | Mock | 0 | 1 | 2 | 0 | 2 | 1 | 0 | 1 | 7 |
| | MRI | 0 | 0 | 0 | 0 | 7 | 3 | 0 | 1 | 11 |
| Missed trials | Mock | 3 | 2 | 1 | 2 | 3 | 2 | 0 | 1 | 14 |
| | MRI | 1 | 3 | 1 | 1 | 3 | 0 | 1 | 0 | 10 |
| Retained trials | Mock | 234 | 232 | 228 | 234 | 226 | 230 | 234 | 231 | 1849 |
| | MRI | 213 | 213 | 219 | 216 | 203 | 213 | 216 | 217 | 1710 |
| Total removed | Mock | 6 | 8 | 12 | 6 | 14 | 10 | 6 | 9 | 71 |
| | MRI | 7 | 7 | 1 | 4 | 17 | 7 | 4 | 3 | 50 |

**Motion assessment (custom controller VS control tasks).** Framewise Displacement (FD) values were obtained from the confounds file generated by the fMRIprep preprocessing pipeline [25] applied to each of the the datasets listed above. For each run of every dataset, the median of FD values were computed, then compared across datasets separately for each subject using two-tailed Kolmogorov-Smirnov tests for independent samples with FDR (Benjamini-Hochberg) correction for multiple comparisons. Specifically, we tested the differences between the *gamepad* dataset and the *friends*, *hcp-rest*, *hcp-gamble* and *hcp-motor* datasets for all 4 subjects, and the differences between the *shinobi* dataset and all others datasets for the 3 subjects that performed the shinobi task.

fMRI temporal SNR (with versus without controller). Temporal Signal-to-Noise Ratio (SNR) maps were computed on each run then averaged per condition (data from the tasks was divided into one of two conditions; with or without the controller present in the scanner) using Nipype [30]. For each voxel, the mean and standard deviation across the distribution of runs were computed and displayed separately for the two conditions. Additionally, we extracted the voxels located within an EPI mask (i.e. voxels located in the gel phantom) and compared the distributions of mean and standard deviation using independent T-Tests.

**MEG noise covariance (with versus without controller).** Noise covariance matrices were computed on each empty room acquisition run using the MNE-python toolbox [31]. Because covariance matrices are symmetric, only the upper part of the matrices were used for subsequent analysis. The covariance matrices of the 20 recordings of each subset (T1, T2, T3 and T4) were averaged separately and then Z-scored. The difference between these two distributions of covariances was assessed using a Wilcoxon signed-rank test. The Pearson correlation between the two distributions of values was computed to estimate the effect size of that difference. Finally, pairwise Pearson correlations were computed between individual recordings of each pair of subsets to compare the distributions of correlation coefficients across subsets.

## Results

### Utility ratings

The answers (N = 4) to each question of the utility ratings survey were averaged and inspected separately. All scales ranged from 1 to 5, although the interpretation of each score varies as

summarized below and detailed in the methods section. For smoothness during operation, the average rating was 4.75 (±0.5 std, higher is better). For the force required for actuation and the operation speed, the average rating was 3.25 (±0.5 std, a score of 3 is best). The mental effort required for operation was rated at 3 (±0 std, a score of 3 is best). Both finger and wrist fatigue were rated below 2 (2±1.15 std and 1.25±0.5 std respectively, lower is better). Finally, both general comfort and overall useability were rated above 4 (4.25±0.5 std and 4.75±0.5 std respectively, higher is better).

### Response latencies (custom controller versus reference controller)

Analysis of reaction times (Fig 5) showed no differences (all p > 0.05, see Table 2) between our MRI-compatible controller (MRI condition) and the commercial controller (Mock condition) when comparing each key separately. Analysis of release reaction times in long trials revealed significant (p < 0.05) differences for 6 of the 8 keys, showing that the MRI-compatible controller led to longer release reaction times (-0.417 < Cohen's D < -0.040 for all significant comparisons) than the reference controller. Analysis of press durations in short trials also showed significant differences (p < 0.05) for all keys, showing that the duration of short presses was longer using our MRI-compatible controller (-1.332 < Cohen's D < -0.300 for all comparisons).

For reference, the naturalistic key durations observed during play in the *shinobi_training* dataset (Table 3) ranged from from an average of 164ms (for phasic actions that aren't impacted by keypress duration, such as hitting an enemy) to 1153ms for tonic actions that can be maintained over time, such as moving to the right.

### Motion assessment (custom controller versus control tasks)

Analysis of run-median FD values (Fig 6 and Table 4) showed no significant differences of motion between the *gamepad* dataset and the other datasets that didn't use our MRI-compatible controller, namely the *hcp-rest*, *hcp-motor* and *hcp-gamble* datasets and the *friends* dataset (all p > 0.05), with the exception of 2 subjects that showed significant differences between the gamepad and *friends* datasets (p < 0.01). For all 3 subjects, the *shinobi* dataset had significantly higher run-median FD values than all the other datasets (all p < 0.05), except for one subject that showed no significant difference between the *shinobi* dataset and the *hcptrt* resting-state dataset.

### fMRI temporal SNR

The temporal SNR maps showed similar distributions of means and standard deviations across the runs acquired with and without the presence of our controller in the MRI scanner (Fig 7). Comparing the average of voxels located within the EPI mask, the two distributions showed significant differences (both p < = 1.00e-04; $M_{controller}$ = 80.88; $STD_{controller}$ = 0.88; $M_{nocontroller}$ = 78.11; $STD_{nocontroller}$ = 0.72) suggesting a slight decrease in noise levels after the controller was introduced in the MRI scanning room.

### MEG noise covariance

Analysis of the difference between averaged covariance matrices acquired before (T3) and after (T4) the controller installation (Fig 8) showed no significant difference (Wilcoxon signed-rank test, p = 0.139). In contrast, although the controller wasn't present in the scanner during recordings of both T1 and T2, the comparison between these two subsets led to a significant difference (Wilcoxon signed-rank test, p = 0.004), showing that the impact of

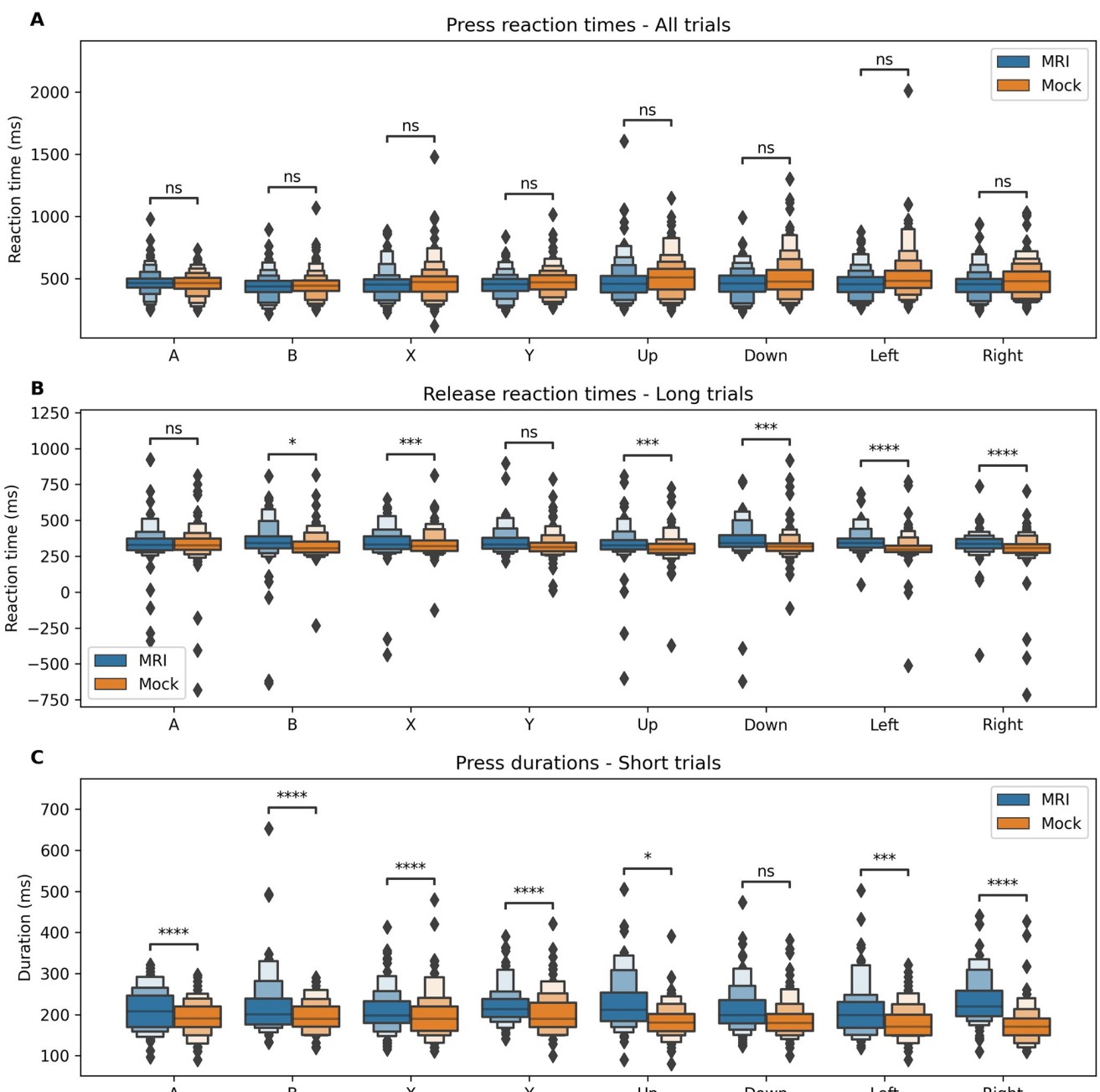

**Fig 5. Distributions of response latencies (pooled across subjects) in the MRI and Mock conditions.** ns: $p <= 1.00e+00$; *: $1.00e-02 < p <= 5.00e-02$; **: $1.00e-03 < p <= 1.00e-02$; ***: $1.00e-04 < p <= 1.00e-03$; ****: $p <= 1.00e-04$. **A** Comparison of reaction times (ms). **B** Comparison of release reaction times (ms) for long trials only. **C** Comparison of total key press durations (ms) for short trials only.

environmental fluctuations on noise levels could be detected by this method, while the controller's effect could not. In both cases (T1 vs T2 and T3 vs T4), the Pearson correlation between the two averaged matrices was fairly high (r = 0.901 and r = 0.953, respectively). Finally, the similarity of the distributions of pairwise correlations between individual recordings in the different subsets further suggests that the noise patterns observed after the

**Table 2. Statistics table showing the comparisons between the MRI and Mock conditions.**

| | | Mock | | | MRI | | | Difference | | |
|---|---|---|---|---|---|---|---|---|---|---|
| | | N | Mean | STD | N | Mean | STD | KS | D | Mean |
| **Press reaction times—All trials (ms)** | A | 234 | 460 | 83 | 213 | 466 | 92 | 0.071 (ns) | -0.071 | 6.251 |
| | B | 232 | 444 | 96 | 213 | 436 | 95 | 0.022 (ns) | 0.084 | 8.012 |
| | Down | 230 | 501 | 155 | 213 | 461 | 121 | 0.000 (ns) | 0.293 | 40.675 |
| | Left | 234 | 514 | 170 | 216 | 457 | 108 | 0.000 (ns) | 0.397 | 56.612 |
| | Right | 231 | 488 | 132 | 217 | 455 | 113 | 0.026 (ns) | 0.266 | 32.773 |
| | Up | 226 | 508 | 141 | 203 | 477 | 156 | 0.015 (ns) | 0.207 | 30.625 |
| | X | 228 | 473 | 142 | 219 | 452 | 109 | 0.020 (ns) | 0.164 | 20.772 |
| | Y | 234 | 472 | 104 | 216 | 449 | 94 | 0.001 (ns) | 0.233 | 23.132 |
| **Release reaction times—Long trials (ms)** | A | 116 | 329 | 162 | 104 | 338 | 153 | 0.100 (ns) | -0.058 | 9.118 |
| | B | 117 | 325 | 101 | 106 | 344 | 182 | 0.283*** | -0.129 | 19.089 |
| | Down | 113 | 330 | 113 | 106 | 360 | 158 | 0.312**** | -0.215 | 29.518 |
| | Left | 114 | 307 | 117 | 107 | 357 | 87 | 0.449**** | -0.49 | 50.538 |
| | Right | 112 | 295 | 152 | 107 | 325 | 127 | 0.284*** | -0.22 | 30.775 |
| | Up | 109 | 309 | 107 | 97 | 331 | 154 | 0.266*** | -0.17 | 22.609 |
| | X | 110 | 338 | 97 | 110 | 343 | 128 | 0.136 (ns) | -0.047 | 5.318 |
| | Y | 116 | 327 | 93 | 107 | 357 | 98 | 0.185* | -0.314 | 30.006 |
| **Press durations—Short trials (ms)** | A | 118 | 192 | 40 | 109 | 211 | 49 | 0.245*** | -0.439 | 19.633 |
| | B | 115 | 195 | 36 | 107 | 222 | 74 | 0.198* | -0.457 | 26.578 |
| | Down | 117 | 189 | 46 | 107 | 213 | 57 | 0.293**** | -0.47 | 24.418 |
| | Left | 120 | 180 | 41 | 109 | 208 | 61 | 0.287**** | -0.536 | 27.904 |
| | Right | 119 | 176 | 48 | 110 | 236 | 62 | 0.580**** | -1.078 | 59.652 |
| | Up | 117 | 186 | 41 | 106 | 231 | 72 | 0.321**** | -0.762 | 44.517 |
| | X | 118 | 201 | 57 | 109 | 209 | 52 | 0.149 (ns) | -0.152 | 8.34 |
| | Y | 118 | 202 | 50 | 109 | 222 | 45 | 0.341**** | -0.412 | 19.595 |

One-tailed (MRI > Mock) Kolmogorov-Smirnov tests for independent samples, with FDR (Benjamini-Hocheberg) correction. KS: Kolmogorov-Smirnov statistics. D: Cohen's D. ns: p < = 1.00e+00;

\*: 1.00e-02 < p < = 5.00e-02;

\*\*: 1.00e-03 < p < = 1.00e-02;

\*\*\*: 1.00e-04 < p < = 1.00e-03;

\*\*\*\*: p < = 1.00e-04.

addition of our custom controller weren't altered more than expected by normal environmental fluctuations.

## Reproduction

A reproduction of the CNeuroMod controller was built by the engineering team at Neurospin and members of the MIND team at Inria (Paris-Saclay, France; Fig 9). The MIND team was not involved in the design process. The reproduced controller was built according to the design plans and documentation made available online and was fitted to their local MRI system. Minor modifications were implemented by the MIND team, for example the shutter and the cap were printed together instead of assembled after printing and the fiber optic cable was 12m long instead of 10m. Other minor assembly issues (buttons getting stuck) were reported and troubleshooted directly with AC, although other teams will also be able to obtain support via GitHub. The controller was tested by 9 different players during gameplay on Super Mario

**Table 3. Descriptive statistics of keypress durations observed in the training phase of the *shinobi* dataset.** These values reflect typical keypress durations when playing an action platformer videogame on a regular controller.

| Subject | Key | Duration (ms) | | | | | | | |
|---|---|---|---|---|---|---|---|---|---|
| | | Count | Mean | STD | Min | 25% | 50% | 75% | Max |
| sub-01 | B (Hit) | 27939 | 134 | 45 | 16 | 116 | 133 | 150 | 1400 |
| | Y (Jump) | 29645 | 350 | 187 | 16 | 200 | 317 | 467 | 1484 |
| | Down | 24710 | 469 | 423 | 16 | 233 | 350 | 567 | 9367 |
| | Left | 10683 | 669 | 670 | 16 | 200 | 450 | 883 | 15134 |
| | Right | 29014 | 1320 | 2123 | 16 | 133 | 567 | 1600 | 38650 |
| | Up | 6478 | 428 | 1055 | 16 | 33 | 66 | 250 | 16217 |
| sub-02 | B (Hit) | 49597 | 235 | 169 | 16 | 117 | 183 | 316 | 3800 |
| | Y (Jump) | 41653 | 426 | 263 | 16 | 250 | 366 | 533 | 2800 |
| | Down | 17697 | 791 | 730 | 16 | 316 | 634 | 1000 | 8967 |
| | Left | 11559 | 616 | 613 | 16 | 200 | 434 | 784 | 13950 |
| | Right | 48203 | 1228 | 1606 | 16 | 100 | 750 | 1766 | 56317 |
| | Up | 15443 | 490 | 511 | 16 | 150 | 367 | 584 | 7316 |
| sub-03 | B (Hit) | 2155 | 180 | 214 | 16 | 133 | 150 | 183 | 4834 |
| | Y (Jump) | 904 | 508 | 384 | 17 | 217 | 384 | 704 | 4700 |
| | Down | 747 | 824 | 858 | 16 | 250 | 667 | 1075 | 13533 |
| | Left | 398 | 575 | 517 | 16 | 233 | 434 | 817 | 4617 |
| | Right | 1627 | 1159 | 1666 | 16 | 300 | 666 | 1358 | 22384 |
| | Up | 328 | 446 | 533 | 16 | 83 | 316 | 621 | 6317 |
| sub-06 | B (Hit) | 25961 | 61 | 53 | 16 | 50 | 50 | 67 | 5516 |
| | Y (Jump) | 14786 | 266 | 260 | 16 | 66 | 150 | 433 | 1483 |
| | Down | 6461 | 473 | 423 | 16 | 200 | 416 | 584 | 7833 |
| | Left | 4699 | 369 | 370 | 16 | 117 | 250 | 508 | 3883 |
| | Right | 17100 | 656 | 897 | 16 | 116 | 334 | 888 | 20017 |
| | Up | 7052 | 361 | 427 | 16 | 96 | 250 | 483 | 6600 |
| Total | B (Hit) | 105652 | 164 | 144 | 16 | 83 | 133 | 183 | 5516 |
| | Y (Jump) | 86988 | 374 | 248 | 16 | 200 | 333 | 500 | 4700 |
| | Down | 49615 | 589 | 581 | 16 | 234 | 433 | 766 | 13533 |
| | Left | 27339 | 593 | 610 | 16 | 183 | 400 | 766 | 15134 |
| | Right | 95944 | 1153 | 1704 | 16 | 117 | 567 | 1550 | 56317 |
| | Up | 29301 | 445 | 658 | 16 | 67 | 283 | 534 | 16217 |

Bros [32] within the MRI scanner and player reported satisfying device useability as measured by the utility ratings. More specifically, the players reported a value of 3 (±0.70 std, a score of 3 is best) for the force required for button activation, 2.55 (±0.88 std, higher is better) for the smoothness during operation, 2.66 (±0.71 std, a score of 3 is best) for the amount of mental effort, 2.88 (±0.33 std, a score of 3 is best) for operation speed, 1.88 (±0.78 std, lower is better) for finger fatigue, 1.33 (±0.71 std, lower is better) for wrist fatigue, 3.33 (±1.22 std, higher is better) for general comfort, and 2.44 (±1.01 std, higher is better) for overall useability.

## Discussion

We built the CNeuroMod videogame controller to be compatible with both fMRI and MEG. We wrote the documentation, plans needed by other teams to reproduce the controller and its

Run-Median of Framewise Displacement values

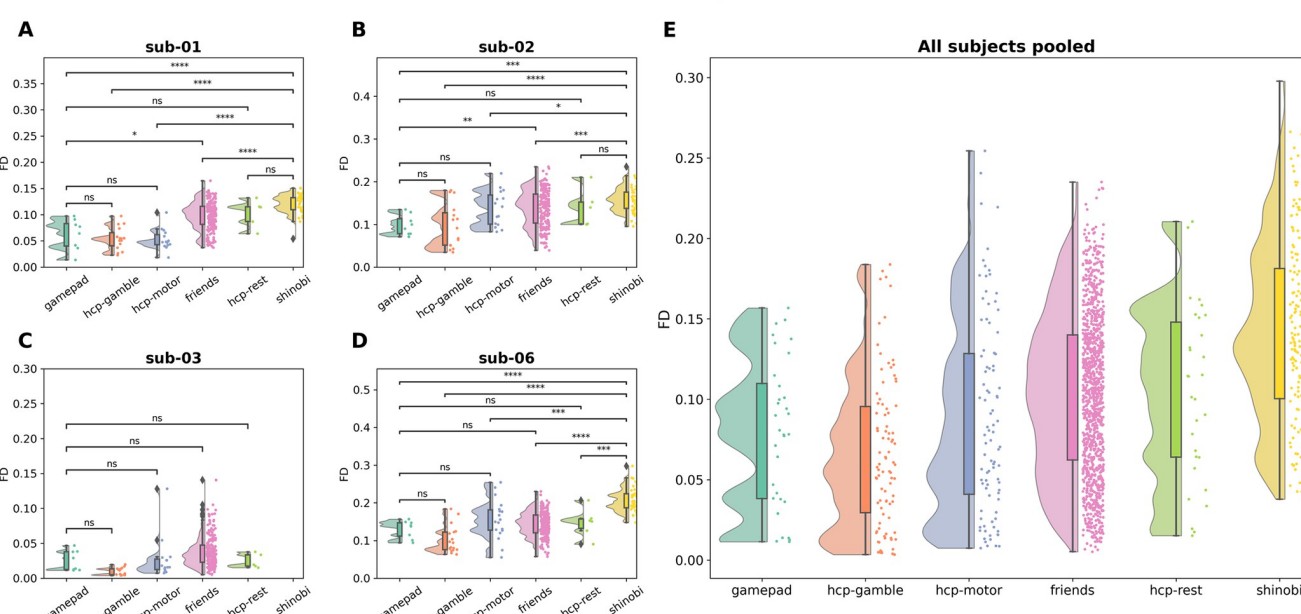

**Fig 6. Distributions of the median of FD values across runs in the datasets included in this study.** Data were analyzed separately for each subject, which showed similar effects (**A-D**). **E** shows the pooled data of all subjects. ns: p < = 1.00e+00; *: 1.00e-02 < p < = 5.00e-02; **: 1.00e-03 < p < = 1.00e-02; ***: 1.00e-04 < p < = 1.00e-03; ****: p < = 1.00e-04.

software and made it publically available under an Open Source Hardware license. A feasibility study showed that our controller consistently registers button presses and releases with only small differences in timing compared with a commercial controller. We showed that the CNeuroMod controller can be used without inducing abnormal motion levels. The presence of the CNeuroMod controller in the scanning environment had small effects on tSNR maps and noise covariance matrices with fMRI and MEG, respectively.

## Open design and reusability

The design files, software and documentation required to build the controller are all released under a CERN-OHL-P license, an Open Hardware License that allows users to reproduce,

**Table 4. Descriptive statistics of the run-median Framewise Displacement (FD) values for the different datasets used in the motion analysis.**

|  | sub-01 | | | | sub-02 | | | | sub-03 | | | | sub-06 | | | | Total | | | |
|---|---|---|---|---|---|---|---|---|---|---|---|---|---|---|---|---|---|---|---|---|
|  | N | Mean | STD | Dur. (hours) | N | Mean | STD | Dur. (hours) | N | Mean | STD | Dur. (hours) | N | Mean | STD | Dur. (hours) | N | Mean | STD | Dur. (hours) |
| friends | 194 | 0.098 | 0.027 | 37.89 | 194 | 0.139 | 0.043 | 37.89 | 196 | 0.039 | 0.022 | 38.308 | 194 | 0.143 | 0.033 | 37.89 | 778 | 0.105 | 0.031 | 151.978 |
| gamepad | 8 | 0.063 | 0.028 | 0.685 | 8 | 0.099 | 0.022 | 0.685 | 8 | 0.025 | 0.014 | 0.685 | 8 | 0.13 | 0.022 | 0.685 | 32 | 0.079 | 0.022 | 2.74 |
| hcp-gamble | 15 | 0.055 | 0.021 | 0.801 | 15 | 0.097 | 0.05 | 0.801 | 15 | 0.01 | 0.005 | 0.801 | 17 | 0.103 | 0.035 | 0.908 | 62 | 0.066 | 0.028 | 3.311 |
| hcp-motor | 15 | 0.053 | 0.019 | 0.901 | 15 | 0.138 | 0.041 | 0.894 | 15 | 0.027 | 0.029 | 0.894 | 18 | 0.156 | 0.051 | 1.073 | 63 | 0.094 | 0.035 | 3.762 |
| hcp-rest | 5 | 0.102 | 0.024 | 1.242 | 5 | 0.141 | 0.041 | 1.242 | 5 | 0.025 | 0.009 | 1.242 | 6 | 0.148 | 0.035 | 1.49 | 21 | 0.104 | 0.027 | 5.216 |
| shinobi | 57 | 0.12 | 0.018 | 10.572 | 56 | 0.158 | 0.029 | 10.735 | 0 | NaN | NaN | 0 | 56 | 0.206 | 0.031 | 11.343 | 169 | 0.161 | 0.026 | 32.65 |

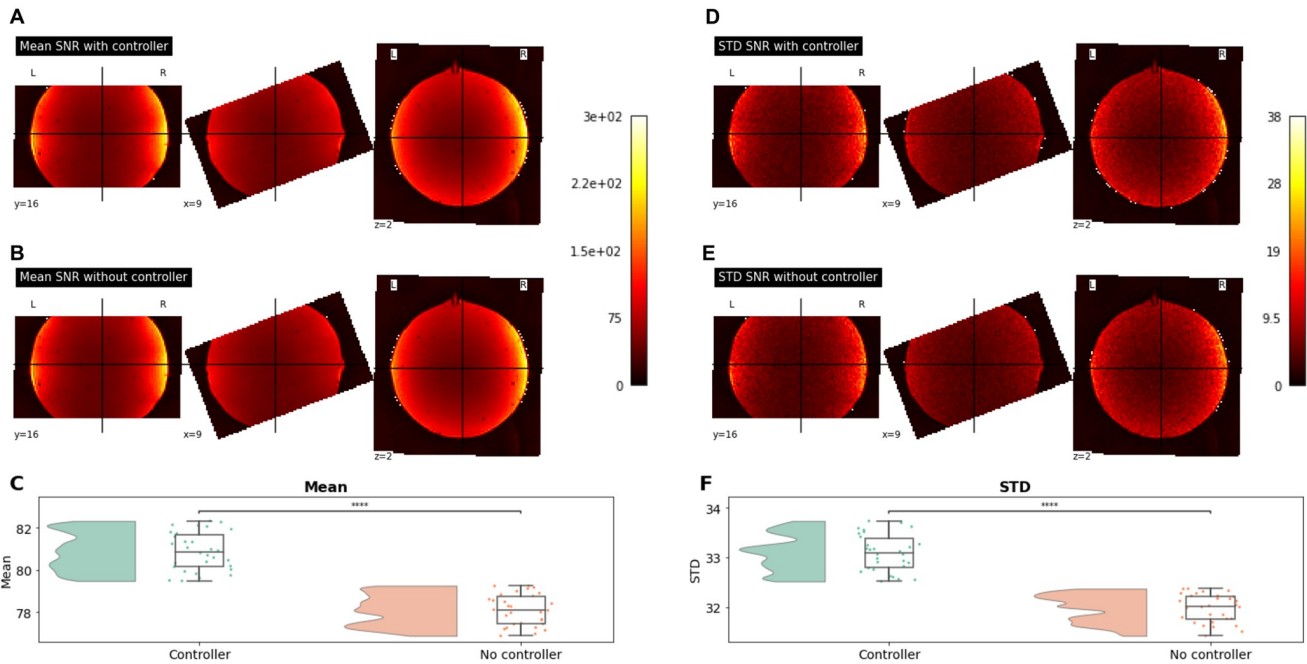

**Fig 7. Temporal SNR maps on runs acquired with and without the controller present in the scanner.** **A** and **B** show the mean SNR across runs recorded with and without the controller respectively. **C** shows the distributions of the averaged SNR values for voxels within the EPI mask. **D**, **E**, and **F** show the same visualizations but for STD values instead. ****: p < = 1.00e-04.

modify and redistribute the covered material. This license has been chosen as it is the most permissive version of the CERN-OHL flavors, with the goal of allowing maximal freedom of reuse to anyone. The presented device requires moderate engineering and technical resources to be built, and the materials used were chosen on the basis of their availability and low costs. All the necessary parts can be obtained for an estimated price around 1000 CAD (Table 5). In comparison, the device proposed by Current Design Inc. (including cables and interface) are sold for a total price of 8390 USD at the time of writing. The provided documentation covers all the steps involved in the preparation, assembly and use of the controller. Through the release of this hardware under a permissive OHL, we anticipate that other research teams or commercial entities will be able to easily reuse and/or modify the controller design beyond purely financial considerations.

## Reactivity of the CNeuroMod videogame controller

The study showed that the CNeuroMod controller consistently registers button presses as well as releases with a temporal precision comparable to a commercial controller. Although no differences were observed in button press reaction times, small significant differences were detected in release reaction times (for long presses) and in press duration (for short presses), showing longer release times with the prototype than with the reference controller. This result might indicate the presence of mechanical friction during the button release, an aspect that could be improved in further design iterations.

This difference in release reaction time could impact game play performance, particularly in games that require dense repetitions of button presses, but is likely negligible when playing most other games, as suggested by the successful collection of the *shinobi* dataset as well as the pilot implemented by the MIND team with *Super Mario Bros [32]*. As a point of reference, the

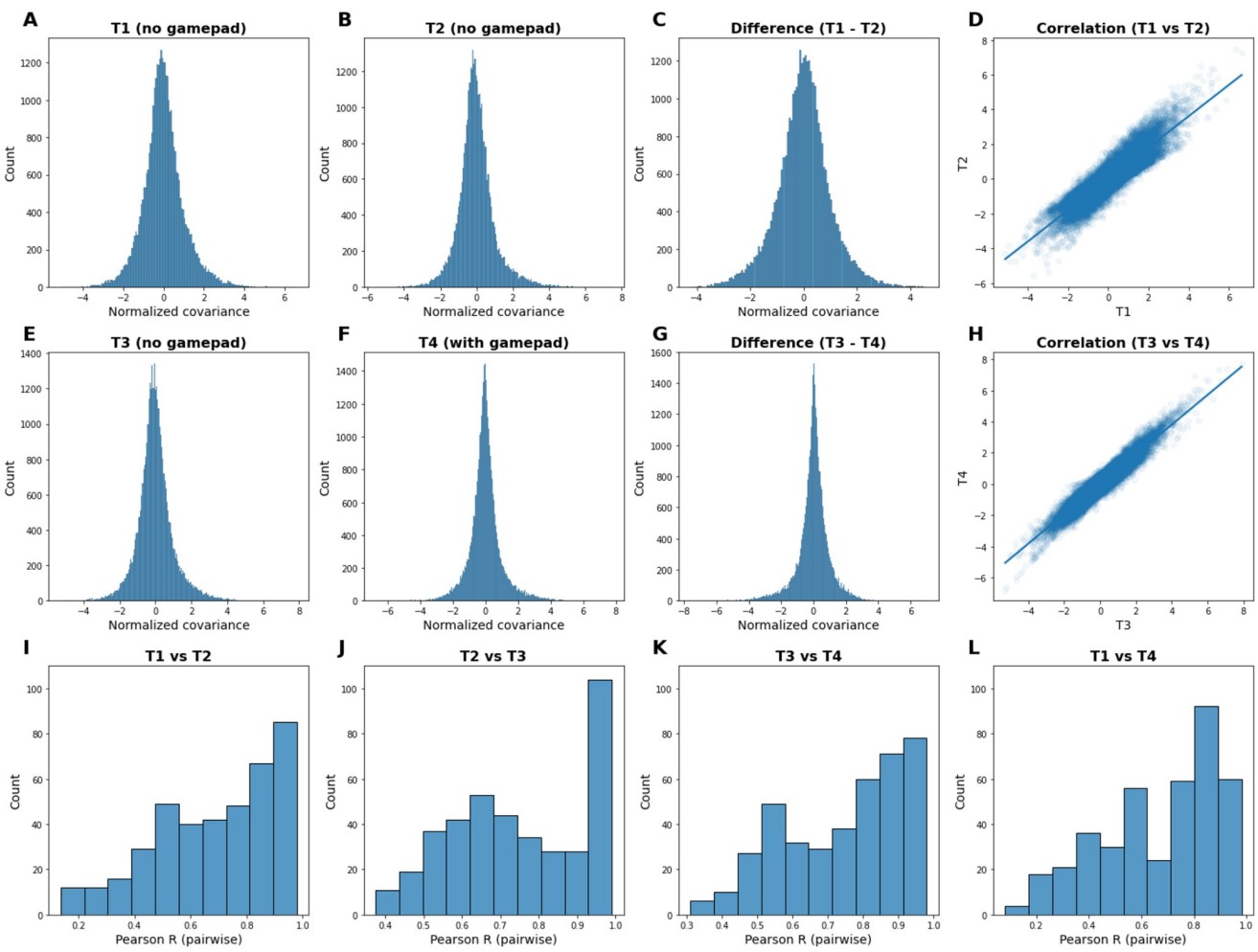

**Fig 8. Comparison of MEG sensors noise covariance over time. A—D** Normalized average covariance across T1 and T2 subsets, and correlation between these two distributions. T1 and T2 both were acquired without the CNeuroMod controller present in the MEG scanner. **E—H** Normalized average covariance across T3 and T4 subsets, and correlation between these two distributions. T3 was acquired without the controller present in the scanner, while T4 was acquired with the controller. **I—L** Distributions of the pairwise correlations between each pair of individual recordings, for each pair of subsets.

difference in short button press duration observed during the *gamepad* task (about 30ms on average) is relatively small compared to the duration of button presses observed while participants played Shinobi using the commercial controller (164ms in average), with very few button presses lasting less than 50ms. In most cases, players can adapt to the controller and learn to release the buttons slightly earlier than they would on another controller. This interpretation of our results was confirmed qualitatively by the participants who all reported that it was necessary to adapt to the controller, but then reported they could play comfortably following this adaptation phase. This effect could also contribute to the utility ratings observed across both instances of the prototype (CNeuroMod and MIND teams). Users from both groups reported great to acceptable usability for all items, although a slightly high finger fatigue (2 for the CNeuroMod sample, 1.88 for the Inria/Neurospin sample) was detected. If the delay at release observed during the cued-response task is caused by mechanical friction, attempts from the subjects at compensating this effect might induce unnecessary finger fatigue, which could be improved in further iterations by adjusting the button and shutter mechanism.

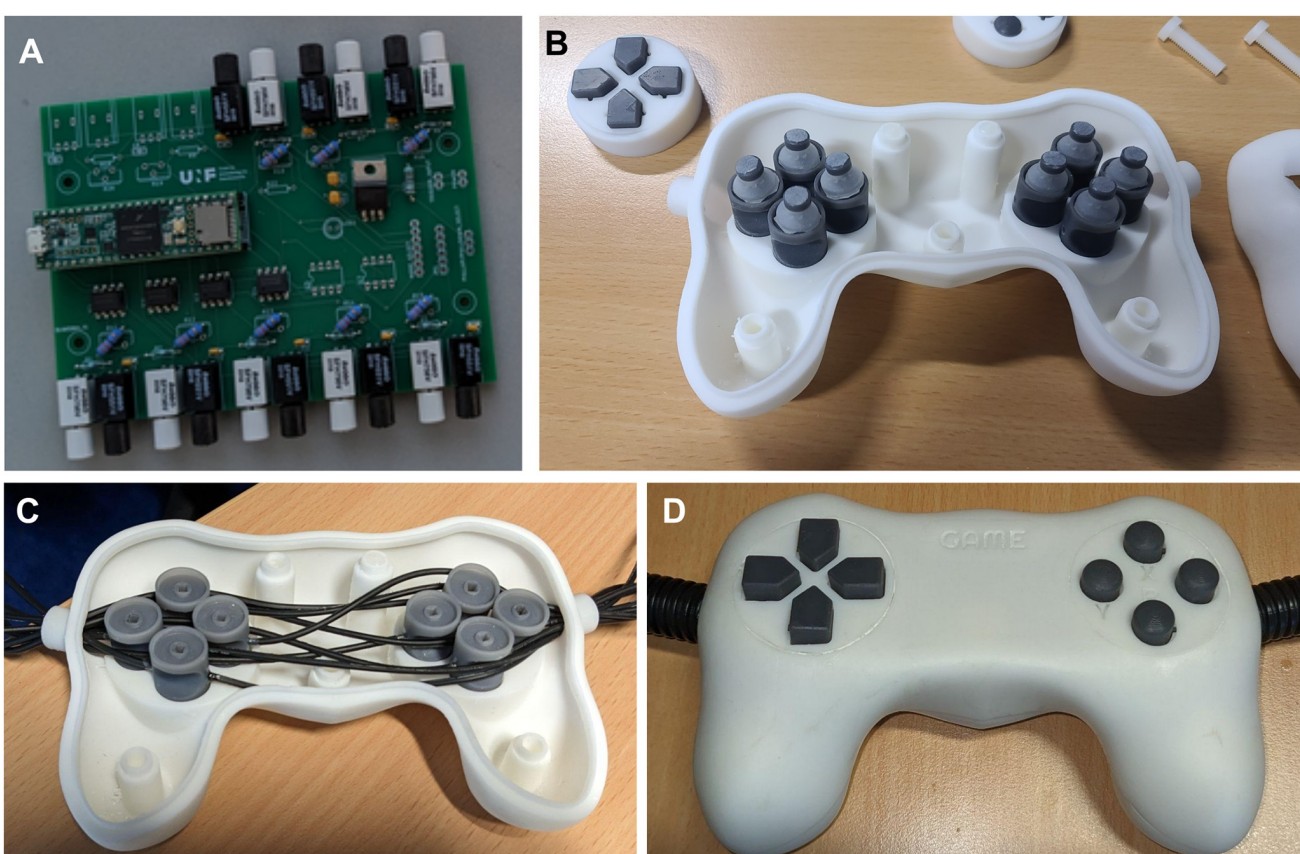

**Fig 9. Pictures of the reproduced controller by team members at Inria/NeuroSpin. A** shows the assembled PCB. **B** shows the bottom part of the controller, with the switch mechanism in the process of being assembled. **C** shows the optics fibers installed in the handheld part. **D** shows the fully assembled controller.

## Motion comparison across various data sets

The analyses conducted on fMRI runs collected on a wide range of tasks showed similar amounts of motion during our *gamepad* task compared to control tasks that did not involve the use of a videogame controller. This suggests that the CNeuroMod controller does not increase the amount of motion in the scanner. Conversely, the comparison between the motion measured during the *shinobi* dataset to that of other control datasets indicated significantly higher levels of motion for the gaming task (*shinobi*), likely to be attributable to the videogame content and not to the controller itself. Even in the case of the *shinobi* dataset, many runs had a fairly low level of motion, with median FD levels below 0.2 or even 0.1. Overall, the usage of the Courtois NeuroMod controller seems compatible with the collection of usable fMRI data, and in some instances even high-quality low-motion data.

## fMRI and MEG compatibility

Analyses performed on fMRI data acquired on a gel phantom head showed a small impact of the CNeuroMod controller on temporal SNR. These differences are statistically significant but are relatively small compared to the amount of noise typically observed during such recordings. Because fMRI recordings are known to be sensitive to thermal noise [33], which can

**Table 5. Bill of material with estimated prices at the time of writing of this manuscript.** More details about the suppliers and item numbers can be found in the documentation.

| Part | Supplier | Qty | Price (CAD) |
|---|---|---|---|
| Fiber optic cable duplex 100m | Digikey | 1 | 279.71 |
| Fiber optic transmitter | Digikey | 8 | 163.2 |
| Fiber optic receiver | Digikey | 8 | 198.72 |
| Teensy 3.5 microcontroller board | Digikey | 1 | 37.76 |
| Gate driver | Digikey | 5 | 7 |
| Hose 13mm ID, black PE 1 | Digikey | 1 | 111.55 |
| Hose 19.48mm ID, black PE 1 | Digikey | 1 | 160.38 |
| Voltage regulator | Digikey | 1 | 2.93 |
| Resistor 330 $\Omega$ | Digikey | 8 | 3.67 |
| Resistor 65 $\Omega$ | Digikey | 8 | 2.39 |
| Capacitor ceramic 0.1 µF | Digikey | 8 | 2.49 |
| Capacitor ceramic 10 µF | Digikey | 2 | 2.3 |
| Connector header pin breakaway 24 pos | Digikey | 4 | 14.28 |
| Connector receptacle 24 pos | Digikey | 2 | 9.18 |
| Polycarbonate Pan Head Philips Screw 8–32 x 1" 2 | McMaster-Carr | 2 | 2 |
| Polycarbonate Pan Head Philips Screw 8–32 x 1/2" 2 | McMaster-Carr | 3 | 3 |
| Polycarbonate Pan Head Philips Screw 4–40 x 3/16" 3 | McMaster-Carr | 8 | 3.49 |
| Nylon 6/6 Female Threaded Round Standoff 8–32 x 1/4" 2 | McMaster-Carr | 5 | 8.35 |
| Nylon 6/6 Female Threaded Round Standoff 4–40 x 1/8" 3 | McMaster-Carr | 8 | 13.12 |
| **Total** | | | **1025.52** |

fluctuate across the time of the day, the observed differences might be attributable to the fact that these two series of recordings were acquired within the same day but at different time and thus, at slightly different room temperatures. Replicating these results with other datasets would further confirm that the amount of noise induced by the controller stays well within acceptable range.

The MEG noise covariance matrices showed no significant impact of the CNeuroMod controller. Further analysis of empty room recordings collected at various periods (from April 2019 to June 2021) showed that the variation of noise levels induced by the controller stayed well within the range of changes that could be expected from normal environmental fluctuations.

## Implications and future work

The CNeuroMod MRI- and MEG-compatible videogame controller provides a versatile device to play most videogames. This versatility extends to any experimental task that requires the use of 8 or less buttons, representing a very large action space. For example, one can use this controller to answer complex questionnaires, e.g. featuring Likert scale, or perform neuropsychological tasks where users navigate many questions. The CNeuroMod controller is specifically geared towards retro gaming from the generation of the Super Nintendo Entertainment System (SNES™, Nintendo). Although some modern titles require basic input sets that can be covered by our controller, most console games published after the release of the XBox™ (Microsoft) and PlayStation™ (Sony) use joysticks and trigger buttons under the index.

Future design iterations could reduce the latency of button releases, and enable the inclusion of additional buttons and analog joysticks that would expand the scope of playable videogames to modern titles.

## Conclusions

The CNeuroMod controller is designed to be built using commonly accessible tools and materials, and released under an Open Hardware License for maximal reusability. A feasibility study showed response latencies comparable to a standard commercial controller. MRI and MEG experiments showed that the controller does not increase subject motion and has marginal effects on acquisition noise, if any. We hope that the CNeuroMod controller will assist researchers in leveraging the richness of videogame tasks in cognitive neuroscience. Our open design can also contribute to a growing ecosystem of open hardware for reproducible neuroimaging research.

## Supporting information

**S1 Appendix. Build instructions and documentation.**
(PDF)

**S1 Dataset. Cued-response task results.**
(CSV)

**S2 Dataset. Shinobi training behavioral data.**
(GZ)

**S3 Dataset. FD values of CNeuroMod datasets.**
(CSV)

**S4 Dataset. fMRI phantom data.**
(CSV)

**S5 Dataset. Noise covariances of MEG empty room data.**
(GZ)

**S6 Dataset. Utility ratings of the controller in both sites.**
(ODS)

**S1 Graphical abstract.**
(TIF)

## Acknowledgments

The authors would like to thank the MEG Imaging Center, the Functional Neuroimaging Unit at CRIUGM and the Courtois NeuroMod scanning crews for acquiring the data.

## Author Contributions

**Conceptualization:** Yann Harel, André Cyr, Julie Boyle, Basile Pinsard, Karim Jerbi, Pierre Bellec.

**Data curation:** Basile Pinsard.

**Formal analysis:** Yann Harel.

**Funding acquisition:** Bertrand Thirion, Karim Jerbi, Pierre Bellec.

**Investigation:** Yann Harel, Pierre Bellec.

**Methodology:** Yann Harel, Julie Boyle, Basile Pinsard.

**Project administration:** Julie Boyle.

**Resources:** André Cyr.

**Software:** Basile Pinsard.

**Supervision:** Karim Jerbi, Pierre Bellec.

**Validation:** Jeremy Bernard, Marie-France Fourcade, Himanshu Aggarwal, Ana Fernanda Ponce, Bertrand Thirion.

**Visualization:** Yann Harel.

**Writing – original draft:** Yann Harel.

**Writing – review & editing:** Yann Harel, André Cyr, Julie Boyle, Basile Pinsard, Jeremy Bernard, Marie-France Fourcade, Himanshu Aggarwal, Ana Fernanda Ponce, Bertrand Thirion, Karim Jerbi, Pierre Bellec.

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
