## [Decision Letter · Decision Letter 0]

19 Jun 2023

PONE-D-23-14328Open design and validation of a reproducible videogame controller for MRI and MEGPLOS ONE

Dear Dr. Harel,

Thank you for submitting your manuscript to PLOS ONE. After careful consideration, we feel that it has merit but does not fully meet PLOS ONE’s publication criteria as it currently stands. Therefore, we invite you to submit a revised version of the manuscript that addresses the points raised during the review process.

We look forward to receiving your revised manuscript.

Kind regards,

Bradley R. King

Academic Editor

PLOS ONE

Journal Requirements:

"The Courtois project on neural modeling was made possible by a generous donation from the Courtois foundation, administered by the Fondation Institut Gériatrie Montréal at CIUSSS du Centre-Sud-de-l’île-de-Montréal and University of Montreal. The Courtois NeuroMod team is based at “Centre de Recherche de l’Institut Universitaire de Gériatrie de Montréal”, with several other institutions involved. See the CNeuromod documentation for an up-to-date list of contributors (https://docs.cneuromod.ca). The replication of the controller by the MIND team, Inria, France has received funding from the European Union’s Horizon 2020 Framework Programme for Research and Innovation under the Specific Grant Agreement No. 945539 (Human Brain Project SGA3) and through the joint Inria “NeuroMind” team grant to PB, KJ, BT and Alexandre Gramfort. PB is a senior fellow (“chercheur boursier senior”) of the “Fonds de recherche du Québec - Santé”. KJ is supported by funding from the Canada Research Chairs program and a Discovery Grant from the Natural Sciences and Engineering Research Council of Canada.

The authors would like to thank the MEG Imaging Center and Courtois-Neuromod scanning crews for acquiring the validation data. "

"The Courtois project on neural modeling was made possible by a generous donation from the Courtois foundation, administered by the Fondation Institut Gériatrie Montréal at CIUSSS du Centre-Sud-de-l’île-de-Montréal and University of Montreal. The Courtois NeuroMod team is based at “Centre de Recherche de l’Institut Universitaire de Gériatrie de Montréal”, with several other institutions involved. See the CNeuromod documentation for an up-to-date list of contributors (https://docs.cneuromod.ca). The replication of the controller by the MIND team, Inria, France has received funding from the European Union’s Horizon 2020 Framework Programme for Research and Innovation (https://research-and-innovation.ec.europa.eu/funding/funding-opportunities/funding-programmes-and-open-calls/horizon-2020_en) under the Specific Grant Agreement No. 945539 (Human Brain Project SGA3) and through the joint Inria “NeuroMind” team grant to PB, KJ, BT and Alexandre Gramfort. PB is a senior fellow (“chercheur boursier senior”) of the “Fonds de recherche du Québec - Santé”. KJ is supported by funding from the Canada Research Chairs (https://www.chairs-chaires.gc.ca/home-accueil-eng.aspx) program and a Discovery Grant from the Natural Sciences and Engineering Research Council of Canada (https://www.nserc-crsng.gc.ca/index_eng.asp). The funders had no role in study design, data collection and analysis, decision to publish, or preparation of the manuscript."

6. We note that Figures 1,2 and 3in your submission contain copyrighted images. All PLOS content is published under the Creative Commons Attribution License (CC BY 4.0), which means that the manuscript, images, and Supporting Information files will be freely available online, and any third party is permitted to access, download, copy, distribute, and use these materials in any way, even commercially, with proper attribution. For more information, see our copyright guidelines: http://journals.plos.org/plosone/s/licenses-and-copyright.

a. You may seek permission from the original copyright holder of Figures 1, 2, and 3 to publish the content specifically under the CC BY 4.0 license. 

Reviewers' comments:

Reviewer's Responses to Questions

**Comments to the Author**

1. Is the manuscript technically sound, and do the data support the conclusions?

Reviewer #1: Yes

Reviewer #2: Yes

2. Has the statistical analysis been performed appropriately and rigorously? 

Reviewer #1: Yes

Reviewer #2: Yes

3. Have the authors made all data underlying the findings in their manuscript fully available?

Reviewer #1: Yes

Reviewer #2: Yes

4. Is the manuscript presented in an intelligible fashion and written in standard English?

Reviewer #1: Yes

Reviewer #2: Yes

5. Review Comments to the Author

Reviewer #1: The manuscript reports results from a project to build and test a prototype handheld videogame controller that is designed for full compatibility with the key technical limitations of brain imaging equipment. The report includes conceptual drawings, programming support and analysis of the prototype’s response characteristics collected from a small sample of participants in the targeted neuroimaging environments.

The motivation, execution and testing analysis of the scanner-compatible controller are explained. The authors’ interpretation of the success of the project is sound and supported by their data, and a confirming example from another research group is reported as well.

The report will interest some neuroimaging research teams that have the skills and mindset to build their own hardware rather than adopt commercially available products. A few concerns are described below, which could be addressed in a revision to make a stronger version of the manuscript.

1. The motivation to build a scanner-compatible videogame controller in-house and publish a thorough engineering analysis of the project is apparent in the Introduction, but it can be stated in more complete fashion.

Based on the prototype, the in-house device costs approximately CAD$1,000 in parts and an undefined amount of experienced labor to build. A bit of investigation reveals that the industry standard Current Designs scanner-compatible videogame controller costs approximately US$4,600, plus fees. The authors appear to justify their project by offering a less costly solution, but a detailed calculation is not presented to support this justification.

A further concern about the relative cost as a justification to build a controller in-house is that the comparability of the haptics and user satisfaction between the device alternatives is not addressed. The participant sample in the results is small and from a narrow age range. Although a larger cohort from a much broader age range would be desirable, some sort of utility rating (as collected from the MIND Team confirmation study) is really needed to fully support the conclusion that the CNeuroMod controller is a suitable substitute for the commercially proven and available alternative.

2. Release reaction times on long trials were significantly longer with the prototype than the mock condition commercial controller (non-scanner compatible). No explanation is offered in the Results or Discussion.

Why is there an apparent difference in the responsiveness of the prototype, depending upon duration of the button presses? How does this responsiveness relate to the poor rating from the MIND Team confirmation study participants (i.e., “a value of 2 for the following items: smoothness during operation and general comfort.”)?

3. The title and text refer to a validation of the original test results, which had showed the comparability of the functional values from the prototype with a non-scanner compatible mock-up device. No true validation study is reported. In other words, there are no results from two or more cohorts performing the same tests as sequential experiments, and there is no analysis of validating results (i.e., an independent samples t-test, for example).

The MIND Team “study” is relevant, but not well described. The measures in the main results do not appear in the MIND Team section (3.4), and the Mind Team user ratings do not appear in the main results. This is not a replication, and referring to the MIND Team information as a replication needs to corrected throughout.

4. The manuscript reads clearly enough, although the voice or writing style changes noticeably between sections of the text. Conforming the entire report to one writing style would improve its readability.

5. Figure 8 includes panels A through L. The legend for Figure 8 only explains panels A through J.

Panels K and L should either be explained in the legend and text, or eliminated from the figure.

Reviewer #2: Overall the paper was quite straightforward and clearly addresses the core question the team started out with. I have one methodological question with regard to how the number of participants and their demographics was chosen. 4 participants isn't very many obviously and the age range of the participants isn't necessarily a good match for what I'd expect of most future work on gaming/neuroimaging. So it'd be useful for the authors to speak to how they determined the sample size and what types of criteria for participants were considered. I'd note though that because I would call this a "feasibility" illustration, the small sample size & potential non-representativeness of the participants aren't "fatal flaws"; but it would be valuable for readers to know how these factors were chosen.

The other issue that I think would improve the paper is a deeper dive into what I would call the exceptionally dated configuration of the controller. The authors indicate that, "Key features of the SNESTM controller can still be found in modern systems such as the XBoxTM (Microsoft) and PlayStationTM (Sony)" which is true, but still reasonably misleading. For instance, it's a bit like if the team created a new car that was basically a Model T Ford, but said that it "still contains many features of modern cars such as the 2023 Honda Accord." It might be true (e.g., it would have 4 wheels, a steering wheel, a windshield), it would still be super dated. Video game controllers have used a dual stick setup since the late 1990s. There are essentially no modern games that I can think of that someone would be able to play with the controller developed here - certainly none of the games used in the research that the authors reference in the introduction (i.e., most of that work on video games/neuroscience has been after the year 2000). So the authors should spend some time on that particular limitation (or else on why it isn't a limitation).

6. PLOS authors have the option to publish the peer review history of their article (what does this mean?). If published, this will include your full peer review and any attached files.

Reviewer #1: No

Reviewer #2: No

---

## [Author Response · Author response to Decision Letter 0]

29 Jul 2023

Response to reviewers

We thank the editor for providing us the opportunity to revise our work, and the reviewers for the insightful comments which have helped improve our study. A point-by-point response to reviewers can be found below. 

Formatting : 

Authors answer : We reformatted the manuscript to follow the PLOS ONE.

Authors answer : The information provided in the Financial Disclosure section of the submission has been re-verified and is correct.

"The Courtois project on neural modeling was made possible by a generous donation from the Courtois foundation, administered by the Fondation Institut Gériatrie Montréal at CIUSSS du Centre-Sud-de-l’île-de-Montréal and University of Montreal. The Courtois NeuroMod team is based at “Centre de Recherche de l’Institut Universitaire de Gériatrie de Montréal”, with several other institutions involved. See the CNeuromod documentation for an up-to-date list of contributors (https://docs.cneuromod.ca). The replication of the controller by the MIND team, Inria, France has received funding from the European Union’s Horizon 2020 Framework Programme for Research and Innovation under the Specific Grant Agreement No. 945539 (Human Brain Project SGA3) and through the joint Inria “NeuroMind” team grant to PB, KJ, BT and Alexandre Gramfort. PB is a senior fellow (“chercheur boursier senior”) of the “Fonds de recherche du Québec - Santé”. KJ is supported by funding from the Canada Research Chairs program and a Discovery Grant from the Natural Sciences and Engineering Research Council of Canada.

The authors would like to thank the MEG Imaging Center and Courtois-Neuromod scanning crews for acquiring the validation data. "

"The Courtois project on neural modeling was made possible by a generous donation from the Courtois foundation, administered by the Fondation Institut Gériatrie Montréal at CIUSSS du Centre-Sud-de-l’île-de-Montréal and University of Montreal. The Courtois NeuroMod team is based at “Centre de Recherche de l’Institut Universitaire de Gériatrie de Montréal”, with several other institutions involved. See the CNeuromod documentation for an up-to-date list of contributors (https://docs.cneuromod.ca). The replication of the controller by the MIND team, Inria, France has received funding from the European Union’s Horizon 2020 Framework Programme for Research and Innovation (https://research-and-innovation.ec.europa.eu/funding/funding-opportunities/funding-programmes-and-open-calls/horizon-2020_en) under the Specific Grant Agreement No. 945539 (Human Brain Project SGA3) and through the joint Inria “NeuroMind” team grant to PB, KJ, BT and Alexandre Gramfort. PB is a senior fellow (“chercheur boursier senior”) of the “Fonds de recherche du Québec - Santé”. KJ is supported by funding from the Canada Research Chairs (https://www.chairs-chaires.gc.ca/home-accueil-eng.aspx) program and a Discovery Grant from the Natural Sciences and Engineering Research Council of Canada (https://www.nserc-crsng.gc.ca/index_eng.asp). The funders had no role in study design, data collection and analysis, decision to publish, or preparation of the manuscript."

Authors answer : We removed the paragraph related to funding information in the “Acknowledgement” section. The information (“Financial Disclosure”) provided in the submission form were verified and are correct.

Authors answer : We saved the cleaned versions of the datasets used to generate the figures as .csv files (and a .pkl file for MEG covariance matrices) and added them to Supporting Information.

Due to restrictions from the ethics committee in charge of the study, human neuroimaging data is only accessible through a registered access model in a databank hosted at our institution. The instructions to apply for access to the data can be found here: https://www.cneuromod.ca/access/access/

Authors answer : We modified the manuscript to add a section named “2.2.2 Ethics statement”, containing the information provided in the submission form.

6. We note that Figures 1,2 and 3in your submission contain copyrighted images. All PLOS content is published under the Creative Commons Attribution License (CC BY 4.0), which means that the manuscript, images, and Supporting Information files will be freely available online, and any third party is permitted to access, download, copy, distribute, and use these materials in any way, even commercially, with proper attribution. For more information, see our copyright guidelines: http://journals.plos.org/plosone/s/licenses-and-copyright.

a. You may seek permission from the original copyright holder of Figures 1, 2, and 3 to publish the content specifically under the CC BY 4.0 license. 

Authors answer : The figure 1 contains only images we created ourselves and for which we own the full rights (both pictures and 3d models). The figures 2 and 3 have been re-created from scratch using only original images for which we own the full rights.

7. Please include captions for your Supporting Information files at the end of your manuscript, and update any in-text citations to match accordingly. Please see our Supporting Information guidelines for more information: http://journals.plos.imagesorg/plosone/s/supporting-information. 

Authors answer : We added a section named Supporting Information that mentions the documentation and dataset files.

Content : 

1a. The motivation to build a scanner-compatible videogame controller in-house and publish a thorough engineering analysis of the project is apparent in the Introduction, but it can be stated in more complete fashion.

Based on the prototype, the in-house device costs approximately CAD$1,000 in parts and an undefined amount of experienced labor to build. A bit of investigation reveals that the industry standard Current Designs scanner-compatible videogame controller costs approximately US$4,600, plus fees. The authors appear to justify their project by offering a less costly solution, but a detailed calculation is not presented to support this justification.

Authors answer : We thank the reviewer for pointing out the need for a more thorough price comparison. We modified the paragraph named “Open design and reusability” to add a table (Table 5) that breaks down the prices of the bill of materials. The Discussion has been modified to also discuss the price of a commercial vs open solution.

1b. A further concern about the relative cost as a justification to build a controller in-house is that the comparability of the haptics and user satisfaction between the device alternatives is not addressed. The participant sample in the results is small and from a narrow age range. Although a larger cohort from a much broader age range would be desirable, some sort of utility rating (as collected from the MIND Team confirmation study) is really needed to fully support the conclusion that the CNeuroMod controller is a suitable substitute for the commercially proven and available alternative.

Authors answer : The utility ratings of the CNeuroMod sample (i.e. the subjects who performed the cued-response task) were added to the paper as well as the answers of the Inria/Neurospin participants. These additional data are now presented in the Utility ratings sections of the Methods and Results. The Discussion has been modified to relate utility ratings to other behavioral results, and the ratings dataset has been added to Supporting Information.

2. Release reaction times on long trials were significantly longer with the prototype than the mock condition commercial controller (non-scanner compatible). No explanation is offered in the Results or Discussion.

Why is there an apparent difference in the responsiveness of the prototype, depending upon duration of the button presses? How does this responsiveness relate to the poor rating from the MIND Team confirmation study participants (i.e., “a value of 2 for the following items: smoothness during operation and general comfort.”)?

Authors answer : We discuss this effect under the section named “Reactivity of the CNeuroMod videogame controller”. We agree that the wording did not make clear that we were addressing specifically the longer release times of our prototype, so we modified the text to make it more explicit. We also added a sentence stating that this effect would likely contribute to the poor utility ratings.

3. The title and text refer to a validation of the original test results, which had showed the comparability of the functional values from the prototype with a non-scanner compatible mock-up device. No true validation study is reported. In other words, there are no results from two or more cohorts performing the same tests as sequential experiments, and there is no analysis of validating results (i.e., an independent samples t-test, for example).

The MIND Team “study” is relevant, but not well described. The measures in the main results do not appear in the MIND Team section (3.4), and the Mind Team user ratings do not appear in the main results. This is not a replication, and referring to the MIND Team information as a replication needs to corrected throughout.

Authors answer : We agree with the reviewer that the terms “replication” and “validation” were used improperly. We removed the term validation from the title. We also replaced references to this study as “validation” for other terms, such as “feasibility study”. We replaced references to “replication” by “reproduction” of the controller device, because the study wasn’t replicated by the MIND team, but they produced a second controller unit.

4. The manuscript reads clearly enough, although the voice or writing style changes noticeably between sections of the text. Conforming the entire report to one writing style would improve its readability.

Authors answer : The style and voice of the manuscript have been reviewed and corrected by one of the authors who is a native English speaker. 

5. Figure 8 includes panels A through L. The legend for Figure 8 only explains panels A through J.

Panels K and L should either be explained in the legend and text, or eliminated from the figure.

Authors answer : The legend was corrected to cover panels A through L.

6. Overall the paper was quite straightforward and clearly addresses the core question the team started out with. I have one methodological question with regard to how the number of participants and their demographics was chosen. 4 participants isn't very many obviously and the age range of the participants isn't necessarily a good match for what I'd expect of most future work on gaming/neuroimaging. So it'd be useful for the authors to speak to how they determined the sample size and what types of criteria for participants were considered. I'd note though that because I would call this a "feasibility" illustration, the small sample size & potential non-representativeness of the participants aren't "fatal flaws"; but it would be valuable for readers to know how these factors were chosen.

Authors answer : We modified the “Participants” section to explain that the subjects were recruited as part of a larger, longitudinal dataset acquisition for which the controller has been developed. We also included a sentence to state that we expected the task to reflect the controllers capability irrespective of videogame experience.

7. The other issue that I think would improve the paper is a deeper dive into what I would call the exceptionally dated configuration of the controller. The authors indicate that, "Key features of the SNESTM controller can still be found in modern systems such as the XBoxTM (Microsoft) and PlayStationTM (Sony)" which is true, but still reasonably misleading. For instance, it's a bit like if the team created a new car that was basically a Model T Ford, but said that it "still contains many features of modern cars such as the 2023 Honda Accord." It might be true (e.g., it would have 4 wheels, a steering wheel, a windshield), it would still be super dated. Video game controllers have used a dual stick setup since the late 1990s. There are essentially no modern games that I can think of that someone would be able to play with the controller developed here - certainly none of the games used in the research that the authors reference in the introduction (i.e., most of that work on video games/neuroscience has been after the year 2000). So the authors should spend some time on that particular limitation (or else on why it isn't a limitation).

Authors answer : We agree that our description of the controllers main features could be misleading. To clarify, we adjusted paragraph 2.1.1. to remove the reference to modern titles and state that the goal behind the design of this controller was to enable retro videogame play, and not necessarily modern titles. We additionally modified the discussion section called “Implications and future work” to explicitly state that future improvements of the design are going to be required to play modern titles.

---

## [Decision Letter · Decision Letter 1]

3 Aug 2023

Open design of a reproducible videogame controller for MRI and MEG

PONE-D-23-14328R1

Dear Dr. Harel,

We’re pleased to inform you that your manuscript has been judged scientifically suitable for publication and will be formally accepted for publication once it meets all outstanding technical requirements.

Kind regards,

Bradley R. King

Academic Editor

PLOS ONE

Additional Editor Comments (optional):

Reviewers' comments:

Reviewer's Responses to Questions

**Comments to the Author**

1. If the authors have adequately addressed your comments raised in a previous round of review and you feel that this manuscript is now acceptable for publication, you may indicate that here to bypass the “Comments to the Author” section, enter your conflict of interest statement in the “Confidential to Editor” section, and submit your "Accept" recommendation.

Reviewer #1: All comments have been addressed

Reviewer #2: All comments have been addressed

2. Is the manuscript technically sound, and do the data support the conclusions?

Reviewer #1: Yes

Reviewer #2: (No Response)

3. Has the statistical analysis been performed appropriately and rigorously? 

Reviewer #1: Yes

Reviewer #2: (No Response)

4. Have the authors made all data underlying the findings in their manuscript fully available?

Reviewer #1: Yes

Reviewer #2: (No Response)

5. Is the manuscript presented in an intelligible fashion and written in standard English?

Reviewer #1: Yes

Reviewer #2: (No Response)

6. Review Comments to the Author

Reviewer #1: (No Response)

Reviewer #2: (No Response)

7. PLOS authors have the option to publish the peer review history of their article (what does this mean?). If published, this will include your full peer review and any attached files.

Reviewer #1: **Yes: **Peter E Wais

Reviewer #2: No

---

## [Editor Report · Acceptance letter]

15 Aug 2023

PONE-D-23-14328R1 

Open design of a reproducible videogame controller for MRI and MEG 

Dear Dr. Harel:

I'm pleased to inform you that your manuscript has been deemed suitable for publication in PLOS ONE. Congratulations! Your manuscript is now with our production department. 

Kind regards, 

on behalf of

Dr. Bradley R. King 

Academic Editor

PLOS ONE